TECHNICAL RELEASE

# SAW: an efficient and accurate data analysis workflow for Stereo-seq spatial transcriptomics

Chun Gong[1,†], Shengkang Li[1,†], Leying Wang[1,†], Fuxiang Zhao[1], Shuangsang Fang[1,2], Dong Yuan[1], Zijian Zhao[1], Qiqi He[1], Mei Li[1], Weiqing Liu[1], Zhaoxun Li[1], Hongqing Xie[1], Sha Liao[1], Ao Chen[1], Yong Zhang[1], Yuxiang Li[1,*] and Xun Xu[3,*]

1 BGI-Shenzhen, Shenzhen, Guangdong, China
2 BGI-Beijing, Beijing, 102601, China
3 BGI-Wuhan, Wuhan, Hubei, China

## ABSTRACT

The basic analysis steps of spatial transcriptomics require obtaining gene expression information from both space and cells. The existing tools for these analyses incur performance issues when dealing with large datasets. These issues involve computationally intensive spatial localization, RNA genome alignment, and excessive memory usage in large chip scenarios. These problems affect the applicability and efficiency of the analysis. Here, a high-performance and accurate spatial transcriptomics data analysis workflow, called Stereo-seq Analysis Workflow (SAW), was developed for the Stereo-seq technology developed at BGI. SAW includes mRNA spatial position reconstruction, genome alignment, gene expression matrix generation, and clustering. The workflow outputs files in a universal format for subsequent personalized analysis. The execution time for the entire analysis is ~148 min with 1 GB reads 1 × 1 cm chip test data, 1.8 times faster than with an unoptimized workflow.

**Submitted:** 29 June 2023

\* Corresponding authors. E-mail: liyuxiang@genomics.cn; xuxun@genomics.cn

† Contributed equally.

Preprint submitted at https://doi.org/10.1101/2023.08.20.554064

Included in the series: *Spatial Omics: Methods and Application* (https://doi.org/10.46471/GIGABYTE_SERIES_0005)

**Subjects** Imaging, Bioinformatics, Cell Biology

## STATEMENT OF NEED

Stereo-seq of BGI STOmics [1] is a panoramic spatial transcriptome technology that achieves ultra-high throughput and ultra-high precision. By capturing mRNA in tissues with the Stereo-seq chip and restoring it to its spatial location, *in situ* sequencing of tissues is achieved, laying the foundation for a deeper understanding of the relationship between gene expression, morphology, and the local environment of cells.

Due to its ultra-high throughput and ultra-high precision, Stereo-Seq generates a large amount of data, which poses a challenge for data analysis. Therefore, efficient analysis tools are needed. In addition, accurate spatial positioning is an important part of data analysis; hence, accurate positioning lays a good foundation for subsequent analysis.

In spatial transcriptomics data analyses, large amounts of data can lead to performance issues in the traditional analysis process. Firstly, the alignment of mRNA sequences, whether using STAR [2] (RRID:SCR_004463) or other software, cannot meet the performance requirements in the current situation. In the s1 (1 cm × 1 cm) chip, this step can account for 70% of the processing time. In addition, the coordinate ID (CID) mapping step is also an essential step in the process, and its accuracy affects the efficiency of spatial positioning.

In this step, CID and coordinates need to be recorded in memory for real-time query and spatial positioning of reads. Faced with large chips, such as S6 (6 cm × 6 cm), the spatial coordinate points can reach as many as 15 billion, and the data structure that stores the correspondence between the CID and the coordinates occupies a lot of memory. Moreover, querying in such a large table can be slow, especially when considering fault tolerance, as the computational complexity and time consumption can increase further. Finally, on large chips, matrix operations of the same size as the chip also have performance bottlenecks, such as excessive memory usage and slow speed. These problems need to be solved through high-performance computing technologies.

We developed the Stereo-Seq Analysis Workflow (SAW), a standard analysis process of Stereo-Seq data. Taking FASTQ [3, 4] as inputs, SAW performs mRNA spatial location restoration, filtering, mRNA genome alignment, gene region annotation, molecule identity (MID) correction, expression matrix generation, tissue region extraction, clustering, saturation analysis, and report generation to obtain the gene expression and spatial information of tissues. Hence, SAW provides the complete basic analyses required for spatial transcriptomics data.

## IMPLEMENTATION

### Processing and parallelization of spatial information in large chips

The principle of spatial localization of sequencing data by spatial transcriptomics is to mark the spatial position and sequencing reads with a 25 bp CID sequence. It then requires locating the sequencing reads back to their original spatial position by matching the CID sequence on the reads. However, as the DNA sequence obtained by current sequencers is not 100% accurate and has a certain error rate, a margin of error tolerance is required when matching the CID sequence. The current error tolerance strategy is to replace each base on the CID sequence with the other three bases (the gene sequence comprises four bases, A, G, C, and T) and then perform the matching.

Due to the high resolution and large field of view of Stereo-seq, the number of spatial coordinate points is very large. For example, for the S6 chip (6 cm × 6 cm), the number of spatial coordinate points can reach as many as 15 billion. Simply storing the corresponding relationship between each coordinate point and the CID sequence in a data structure can consume more than 600 GB of memory, and the query speed can be very slow, seriously affecting the applicability and analysis efficiency of standard analyses.

Therefore, we split the spatial coordinates and CID information, which are stored in a mask file. Correspondingly, the FASTQ files are also split according to the same rules. For example, if the mask file needs to be split into four parts, the first base of the CID sequence can be used as the classification criterion and split into four parts starting with A, C, G, and T. If 16 parts are needed, the first two bases can be used as the classification criterion. If a non-power-of-4 number of parts is needed, such as ten parts, we can use a modulo operation. Similarly, the FASTQ file can be split according to the CID sequence using the same rule, and the corresponding mask and FASTQ files belonging to the same category need to undergo CID mapping. This step solves the above memory problem and improves the parallelization of data processing (Figure 1). The mask file (which records the corresponding relationship between the CID sequence and the spatial coordinates) and the FASTQ split are paired for subsequent analyses, and then merged when needed.

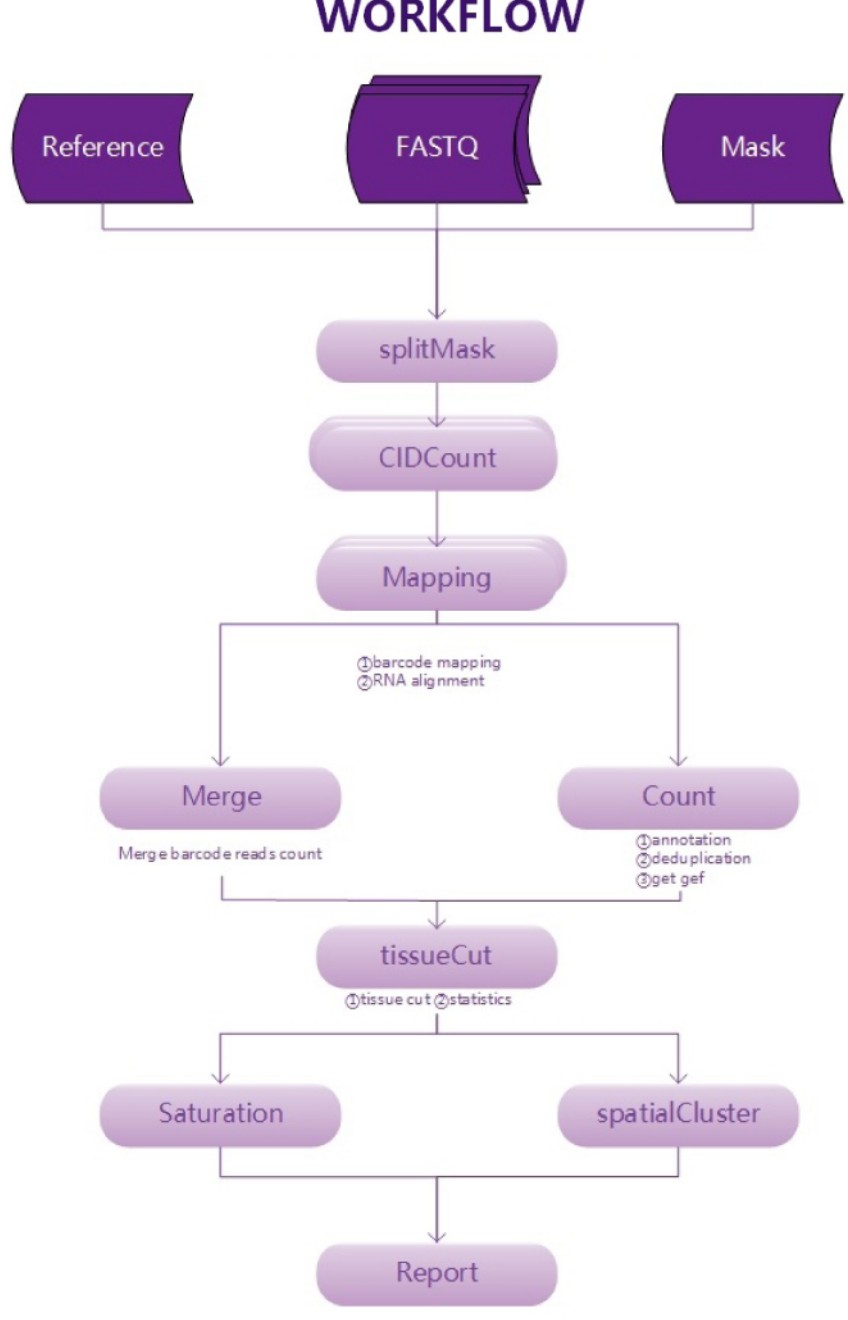

**Figure 1.** Stereo-seq data analysis workflow.

## Rapid alignment of genomes

The successful positioning of spatial location in read goes through several filtering steps, including whether MID contains N bases, whether MID is poly-A, the MID quality value, and whether mRNA contains poly-A. Reads filtered through these steps undergo genome alignment and output a BAM [5] file containing alignment information. Currently, commonly used RNA alignment tools include STAR, Hisat2 [6] (RRID:SCR_015530), and

TopHat2 [7] (RRID:SCR_013035). Among them, STAR is known for its high unique alignment rate and relatively fast speed, but it still cannot meet the requirements of sizeable spatial omics datasets. Therefore, we made a series of optimization attempts, including using efficient multi-threaded input-output (IO) models, single instruction multiple data, improving L2 cache hit rate and other micro-architectural optimization techniques, redesigning business-level algorithms in data processing, and using FM-index technology in maximum matching prefix search, ultimately accelerating it by two times.

### Gene expression matrix generation

Gene expression quantification analysis of STOmics is achieved through the count tool in the analysis software. Count annotates uniquely mapped reads based on mapping alignment results, combined with the reference gene annotation file (GFF/GTF [8, 9]) of the corresponding species, and then corrects and deduplicates MID, generating processed BAM files, gene annotation, and MID correction and deduplication.

#### *Gene region annotation process*

(1)  For each read, search for overlap with the gene interval in the annotation file, calculate the gene name/strand on the annotation, determine whether it is EXONIC, INTRONIC, or INTERGENIC, and count how many belong to each type.

(2)  Parse the cigar information. Obtain the entire length of the read and the starting position and length of each align block.

(3)  Search for all overlapping genes.

(4)  Determine which gene to choose. For each gene:

  (a)  For each align block, calculate with each transcript of the current gene, obtain multiple exoncnt/introncnt based on the length of overlap with exon/non-exon regions.

  (b)  Accumulate the cnts of each align block to obtain the optimal exoncnt/introncnt. If the exoncnt is greater than or equal to 50% of the read length, mark it as EXONIC. Otherwise, if the introncnt is greater than or equal to 50% of the read length, mark it as INTRONIC. Otherwise, mark it as INTERGENIC.

  (c)  Choose the most reliable gene from multiple genes.

   (i)  First, obtain a list of genes with the best annotation results (priority is given to EXONIC>INTRONIC>INTERGENIC).

   (ii)  From these genes, select the gene with the most significant overlap as the annotation result.

   (iii)  If multiple genes have the same overlap length, randomly select one gene (the selection rule is to choose the gene with the smallest start and end).

#### *MID correction*

The following process corrects the error MID caused by sequencing errors based on the Hamming distance:

(1)  Data preparation. A nested map of the form {cidgene: {mid: cnt}} stores the number of each CID and gene combination under various MIDs.



(2)  Correction.

(a)  Set parameters: minimum number of MID types threshold/tolerance number threshold/MID length. Default: 5/1/10.

(b)  Correction. For each group of data in the nested map:

(i)  Check the number of MID types and continue processing if it is greater than or equal to the threshold.

(ii)  Sort in descending order according to the cnt of MID, obtaining a list in the form of [(mid1, cnt1), (mid2, cnt2), ...].

(iii)  Traverse the sorted list in reverse order, starting with the MID with the smallest cnt, and calculate the base error with other MIDs. If it satisfies the tolerance number threshold, correct the current MID to the MID with a larger cnt, and transfer the cnt of the current MID to the correct MID.

(iv)  Obtain the nested map after correction in the form of {cidgene: {oldmid: newmid}}.

Expression matrix

(1)  Select reads annotated to EXON or INTRON.

(2)  Filter out reads with directions opposite to the annotated gene chain direction.

(3)  Group by coordinate, gene, and MID in order.

(4)  Count the number of unique MIDs for each coordinate and gene, which is the expression matrix.

(3)  Example

Given a fault tolerance threshold of 1, assuming the sorted MID sequence and count are as follows:

```
{
"AAA": 5,
"GGA": 4,
"AGA": 3,
"AAT": 2,
"GGG": 1,
"CCC": 1
}
```

Correction process:

Calculate the fault tolerance count between `"CCC"` and `"AAA"` - `"GGG"`, all of which are greater than the threshold of 1.

Calculate the fault tolerance count between `"GGG"` and `"AAA"` - `"AAT"`, and find that when encountering `"GGA"`, the fault tolerance count is 1. Then, update the count values of both and record the corresponding relationship before and after correction.

Calculate the fault tolerance count between `"AAT"` and `"AAA"` - `"AGA"`, and find that when encountering `"AAA"`, the fault tolerance count is 1. Then, update the count values of both and record the corresponding relationship before and after correction.

Calculate the fault tolerance count between `"AGA"` and `"AAA"` - `"GGA"`, and find that when encountering `"AAA"`, the fault tolerance count is 1. Then update the count values of both and record the corresponding relationship before and after correction.

Calculate the fault tolerance count between `"GGA"` and `"AAA"`, which is greater than the threshold of 1.

Update the original data to:

```
{
"AAA": 10,
"GGA": 5,
"AGA": 0,
"AAT": 0,
"GGG": 0,
"CCC": 1
}
```

Save the mapping relationship before and after correction:

```
{
"AGA": "AAA",
"AAT": "AAA",
"GGG": "GGA"
}
```

## Extracting data of tissue coverage area

Extracting the data of tissue coverage area is based on the tissue outline. Tissuecut implements two deep learning methods and traditional image processing, compatible with two types of images (i.e., tissue microscopic images and gene expression heatmaps) and designs an end-to-end tissue region extraction algorithm. The deep learning method uses the BiSeNet [10, 11] network algorithm of the neural network algorithm to train two lightweight real-time segmentation network models, which are used to extract tissue regions in microscopic images and gene expression heatmaps. The traditional image processing algorithm mainly extracts tissue regions based on the grayscale value information of the image. The algorithm process is as follows:

For different types of images to be extracted and different algorithms, different image pre-processing processes are used;

Deep learning or traditional algorithms are used to extract tissue regions;

The algorithm results are further processed, noise is filtered, and the final tissue region is obtained. Then, the tissue outline is extracted based on the tissue region, and the data corresponding to the contour coordinates in the region are obtained.

## Clustering

The clustering process for identifying heterogeneity and similarity among cells in tissue regions uses spatial information and gene expression levels. The clustering process involves three steps:

(1) Data preprocessing. This step involves filtering, normalization, and standardization of the data. The purpose of filtering is to remove cells with too few genes. Normalization and standardization aim to transform the data into the same scale and eliminate the adverse effects of outliers.

(2) Feature selection. Principal component analysis and Umap [12] (RRID:SCR_018217) are used for feature selection and dimensionality reduction. The most representative genes are selected from all gene expression values for the subsequent clustering analysis.

(3) Clustering analysis. An unsupervised clustering is performed using the Leiden [13] algorithm, a graph-based clustering method. First, a neighborhood graph is constructed based on the similarity between cells, where each cell is considered a node and the connections between nodes represent their similarity. Each node is initially considered a separate cluster, and the modularity of the entire graph is calculated. Then, adjacent nodes are iteratively merged to improve modularity. In each iteration, the modularity of merging each node with its neighboring nodes is calculated, and the merging method with the highest modularity is selected. When the iterative process converges, the final clustering result is reached.

## Saturation analysis

### Preparation

The formula for calculating the saturation value is 1-(uniq reads/total reads). First, 5% of bin200 unique coordinates are sampled and restored to bin1 coordinates. Then, the sampled bin1 coordinates under tissue are used to filter data, a list of ($x$, $y$, gene, MID) is constructed, and all count values to obtain anno reads are accumulated.

### Saturation calculation

Shuffle the previous list, process the data in order according to the sampling interval of $\{0.05, 0.1, 0.2, 0.3, 0.4, 0.5, 0.6, 0.7, 0.8, 0.9, 1.0\}$, and, for each sampling point, calculate the total reads/saturation value/median number of genes under bin1/bin200, respectively, and output the statistical result file.

Saturation value. Calculate the uniq reads (i.e., gene and MID are both uniq) and the total reads under all bins using the formula 1-(uniq reads/total reads). The algorithms for bin1 and bin200 are the same.

Median number of genes. Calculate the number of uniq genes under each bin and take the median. The algorithms for bin1 and bin200 are the same.

Uniq reads. The uniq reads of bin1 are all the uniq reads (i.e., gene and MID are both uniq) under all bin1; the uniq reads of bin200 are all the uniq reads under all bin200 ($x$, $y$ coordinates of bin1, gene, and MID are all uniq).

## Memory issues in large chip scenarios

In large chip scenarios, in addition to the mask file, storing gene expression information in large IO files and matrix calculations can cause excessive memory consumption. To solve this problem, we used a series of optimization techniques, including batch processing of large IO files and partial matrix calculations, pre-calculating sizes to avoid using dynamically expanding data structures, and designing more finely-tuned custom data-types with smaller memory overhead based on business characteristics. These techniques enabled large chip data to be successfully completed on ordinary memory machines (256 GB).

## EXAMPLES

## mRNA spatial position restoration, filtering, and genome alignment statistics

Taking SS200000135TLD1 data (https://github.com/BGIResearch/SAW/tree/main/testdata) as an example, we executed mRNA spatial position restoration, filtering, and genome

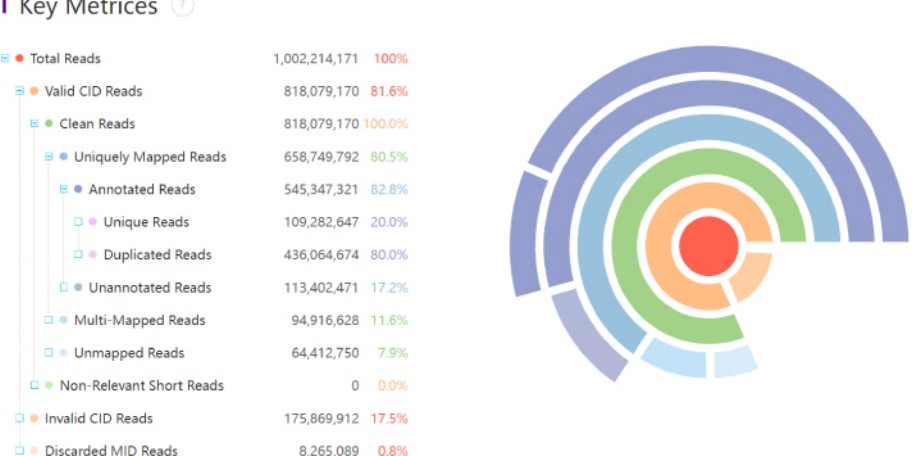

**Figure 2.** Summary of spatial position restoration, filtering, and genome alignment.

**Table 1.** Demo of the reads annotation statistics.

| Type | Number |
|---|---|
| Exonic | 495,050,694 |
| Intronic | 50,296,627 |
| Intergenic | 113,402,471 |
| Transcriptome | 545,347,321 |
| Antisense | 112,611,349 |

alignment in sequence, and obtained the statistics shown in Figure 2. Among 1 GB reads, 818 MB (78.8%, compared to the previous step) could be aligned back to spatial positions. After filtering, 763 MB (93.3%) reads were still left. After alignment to the genome, we obtained 641 MB (84.0%) of uniquely mapped reads.

## Gene expression spatial distribution map
The BAM file generated after genome alignment was annotated and MID-corrected to produce expression information and statistical results. The expression information was stored in hdf5 format and could be visualized (Figure 3). The statistical results provided the number of exonic regions, introns, and intergenic regions annotated. A total of 481 MB reads were annotated in the exonic region (Table 1).

## Spatial clustering
Gene expression profiles were calculated for each position within bin200 (a 200 × 200 grid of points), then spatial clustering was performed (Figure 4). This resulted in 21 classifications, which roughly aligned with the cell clustering results.

## Saturation analysis
Our saturation analysis showed that the median sequencing depth and number of gene species tended to saturate, while the number of unique reads did not yet saturate (Figure 5).

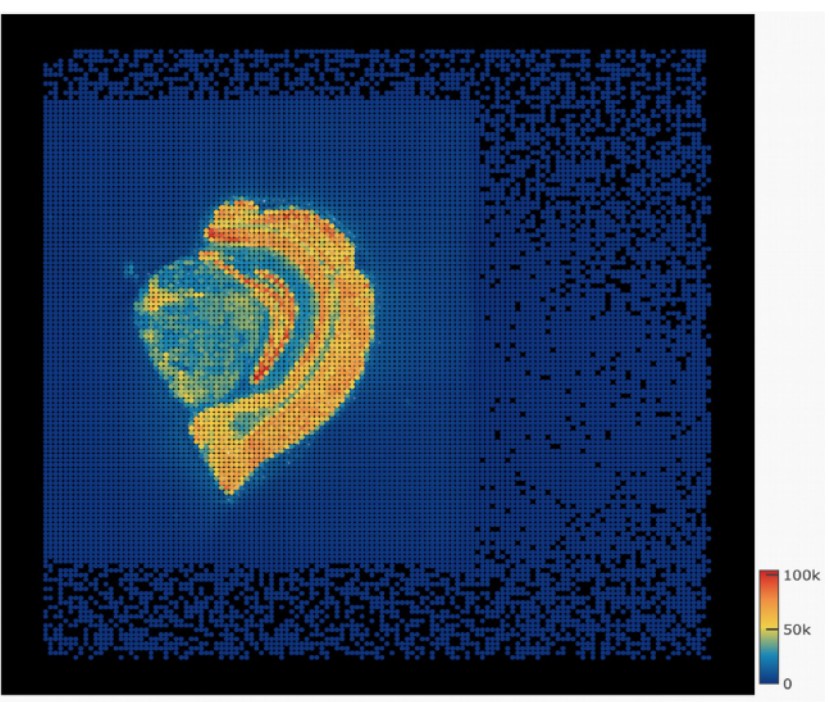

**Figure 3.** A demo of the spatial visualization of gene expression information.

**Table 2.** The elapsed time used by the analysis workflow before and after optimization.

| Performance | Data1 (s1 chip, ~1 GB reads) | | Data2 (s1 chip, ~ 1 GB reads) | |
|---|---|---|---|---|
| | Before optimization | After optimization | Before optimization | After optimization |
| Elapsed time (min) | 263.1 | 148.1 | 227.9 | 127.3 |

## Testing

Each pipeline tool was optimized through a series of high-performance computing techniques. We conducted performance tests on three samples to evaluate changes in runtime and memory. Data1 and data2 are both s1 chips (1 cm × 1 cm) with around 1 billion reads, while data3 is a large chip (2 cm × 3 cm). After optimization, the runtime on data1 decreased from 263.1 to 148.1 min, resulting in a 1.8× speed increase, and the mapping time decreased from 175.0 to 106.9 min. On data2, the runtime decreased from 227.9 to 127.3 min, resulting in a 1.8× speed increase, and the time of mapping decreased from 158.0 to 83.7 min (Table 2). In terms of memory optimization, after process optimization, the memory peak of tissueCut on data3 decreased from well over 83.5 GB to 33.5 GB.

## FUTURE DIRECTIONS

The alignment rate of CID affects the amount of data that enters the subsequent analysis. With our test sample, we obtained a successful alignment rate of 78.8% reads. Due to sequencing errors and alignment algorithm limitations, approximately 20% of reads could not be aligned. In the future, more accurate algorithms (such as those that consider mask CID and fastq CID-mismatch base quality-values and spatial features) or deep learning models may further improve the accuracy of the pipeline.

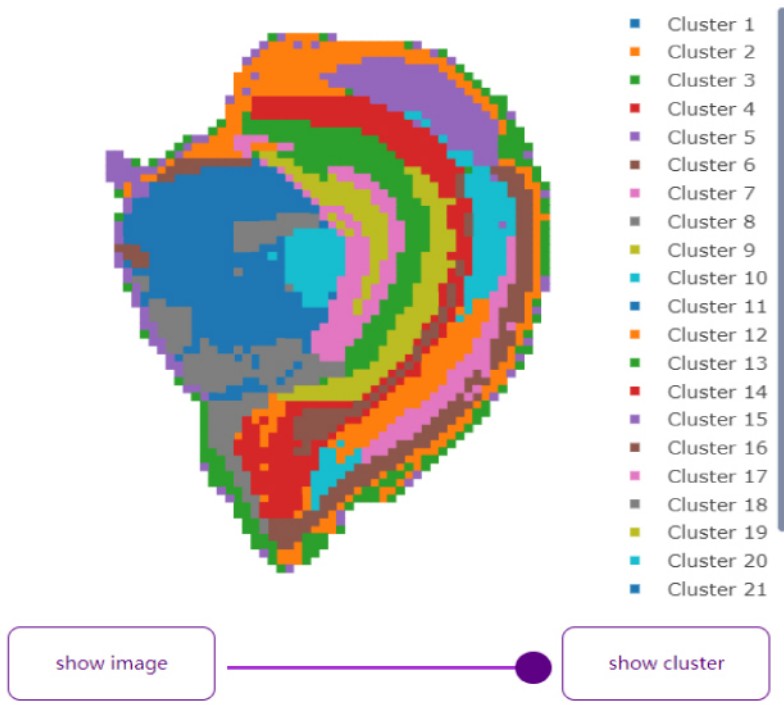

Tissue Plot with Spots (bin200)

**Figure 4.** A demo of the spatial clustering for a mouse brain dataset.

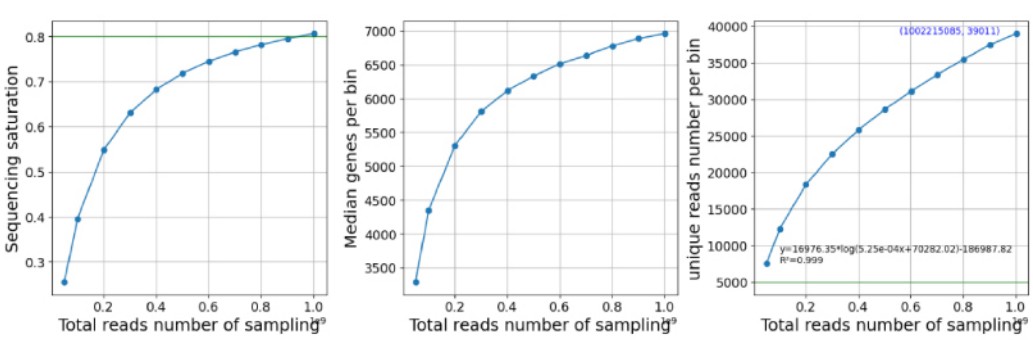

**Figure 5.** A demo of reads saturation statistics.

## AVAILABILITY OF SOURCE CODE AND REQUIREMENTS

- Project name: SAW
- Project home page: https://github.com/BGIResearch/saw_tools (tools source code), https://github.com/STOmics/SAW (script and docker)
- Operating system(s): Linux
- Programming language: C++, Python
- Other requirements: Python >=3.8
- License: GNU General Public License version 3
- RRID:SCR_025001

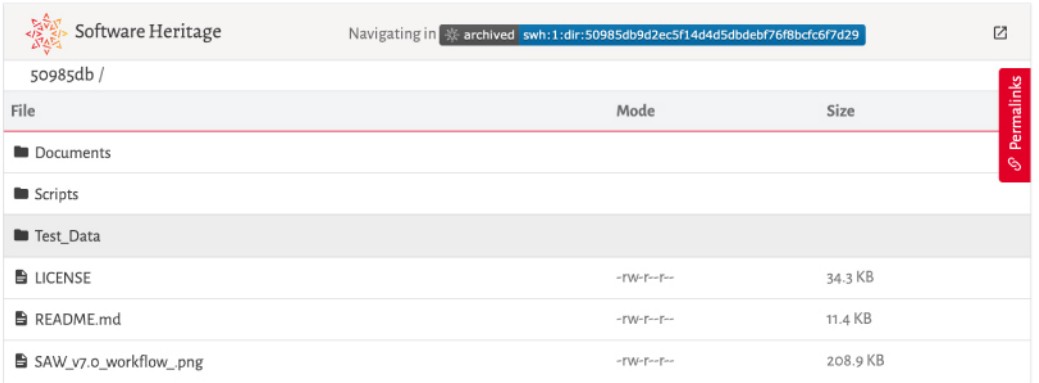

**Figure 6.** Software Heritage archive of the code [18]. https://archive.softwareheritage.org/browse/embed/swh:1:dir:50985db9d2ec5f14d4d5dbdebf76f8bcfc6f7d29;origin=https://github.com/STOmics/SAW;visit=swh:1:snp:6834a6db518ff7d000ce6ffea0cca9956a8347d2;anchor=swh:1:rev:411bab897e0d4642715f1f7b60780b545fb21d12/

## DATA AVAILABILITY

The data supporting this study's findings have been deposited into the CNGB Sequence Archive (CNSA) [14] of the China National GeneBank DataBase (CNGBdb) [15] with the accession number CNP0004437. The raw sequencing data is available in the SRA via BioProject: PRJNA1036005. The test data is in GitHub [16], and additional supporting data is in the GigaDB repository [17]. Archival snapshots of the code are also available in Software Heritage (Figure 6) [18].

## ABBREVIATIONS

CID, coordinate ID; IO, input-output; MID, molecule identity; SAW, Stereo-Seq Analysis Workflow; STOmics, Spatial-Temporal Omics.

## DECLARATIONS

### Ethics approval and consent to participate

The authors declare that ethical approval was not required for this type of research.

### Competing interests

The authors are all employees of BGI.

### Funding

National Key R&D Program of China (2022YFC3400400).

### Acknowledgements

The authors would like to acknowledge STOmic Cloud (https://cloud.stomics.tech) for supplying software analysis, China National GeneBank.

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
