## [Editor Report]

Editor’s AssessmentOne limiting factor in the adoption of spatial omics research are workflow systems for data preprocessing, and to address these authors developed the SAW tool to process Stereo-seq data. The analysis steps of spatial transcriptomics involve obtaining gene expression information from space and cells. Existing tools face issues with large data sets, such as intensive spatial localization, RNA alignment, and excessive memory usage. These issues affect the process's applicability and efficiency. To address this, this paper presents a high-performance open-source workflow called SAW for Stereo-Seq. This includes mRNA position reconstruction, genome alignment, matrix generation, clustering, and result file generation for personalized analysis. During review the authors have added examples of MID correction in the article to make the process easier to understand. And In the future, more accurate algorithms or deep learning models may further improve the accuracy of this pipeline.

---

## [Reviewer Report]

Reviewer name and names of any other individual's who aided in reviewerYanjieWeiDo you understand and agree to our policy of having open and named reviews, and having your review included with the published manuscript. (If no, please inform the editor that you cannot review this manuscript.)YesIs the language of sufficient quality?YesPlease add additional comments on language quality to clarify if neededIs there a clear statement of need explaining what problems the software is designed to solve and who the target audience is? YesAdditional CommentsIs the source code available, and has an appropriate Open Source Initiative license <a href="https://opensource.org/licenses" target="_blank">(https://opensource.org/licenses)</a> been assigned to the code?YesAdditional CommentsAs Open Source Software are there guidelines on how to contribute, report issues or seek support on the code?YesAdditional CommentsIs the code executable?YesAdditional CommentsIs installation/deployment sufficiently outlined in the paper and documentation, and does it proceed as outlined?YesAdditional CommentsIs the documentation provided clear and user friendly?YesAdditional CommentsAdditional CommentsIs there a clearly-stated list of dependencies, and is the core functionality of the software documented to a satisfactory level?YesAdditional CommentsHave any claims of performance been sufficiently tested and compared to other commonly-used packages? YesAdditional CommentsAdditional CommentsAre there (ideally real world) examples demonstrating use of the software? YesAdditional CommentsAdditional CommentsAny Additional Overall Comments to the AuthorRecommendationMinor Revisions

---

## [Reviewer Report]

Reviewer name and names of any other individual's who aided in reviewerZexuan ZhuDo you understand and agree to our policy of having open and named reviews, and having your review included with the published manuscript. (If no, please inform the editor that you cannot review this manuscript.)YesIs the language of sufficient quality?YesPlease add additional comments on language quality to clarify if neededIs there a clear statement of need explaining what problems the software is designed to solve and who the target audience is? YesAdditional CommentsIs the source code available, and has an appropriate Open Source Initiative license <a href="https://opensource.org/licenses" target="_blank">(https://opensource.org/licenses)</a> been assigned to the code?YesAdditional CommentsAs Open Source Software are there guidelines on how to contribute, report issues or seek support on the code?YesAdditional CommentsIs the code executable?YesAdditional CommentsIs installation/deployment sufficiently outlined in the paper and documentation, and does it proceed as outlined?YesAdditional CommentsIs the documentation provided clear and user friendly?YesAdditional CommentsIs there enough clear information in the documentation to install, run and test this tool, including information on where to seek help if required?YesAdditional CommentsIs there a clearly-stated list of dependencies, and is the core functionality of the software documented to a satisfactory level?YesAdditional CommentsHave any claims of performance been sufficiently tested and compared to other commonly-used packages? YesAdditional CommentsIs test data available, either included with the submission or openly available via cited third party sources (e.g. accession numbers, data DOIs)?YesAdditional CommentsAre there (ideally real world) examples demonstrating use of the software? YesAdditional CommentsIs automated testing used or are there manual steps described so that the functionality of the software can be verified?YesAdditional CommentsAny Additional Overall Comments to the AuthorIt would be helpful if some examples can be provided to illustrate the key steps, e.g., the gene region annotation process and MID correction. Some information of the references is missing. Please carefully check the format of the references.RecommendationMinor Revisions